# A Critical-Systematic Review of the Interactions of Biochar with Soils and the Observable Outcomes

Jackson Nkoh Nkoh [1,2,*] , M. Abdulaha-Al Baquy [3,*] , Shamim Mia [4,*] , Renyong Shi [1,2],
Muhammad Aqeel Kamran [5], Khalid Mehmood [6] and Renkou Xu [1,2]

1   State Key Laboratory of Soil and Sustainable Agriculture, Institute of Soil Science, Chinese Academy of
    Sciences, P.O. Box 821, Nanjing 210008, China; ryshi@issas.ac.cn (R.S.); rkxu@issas.ac.cn (R.X.)
2   College of Advanced Agricultural Sciences, University of Chinese Academy of Sciences, Beijing 100049, China
3   Department of Soil Science, Faculty of Agriculture, Hajee Mohammad Danesh Science and Technology
    University, Dinajpur 5200, Bangladesh
4   Department of Agronomy, Patuakhali Science and Technology University, Dumki 8602, Bangladesh
5   College of Environmental and Resource Sciences, Zhejiang University, Hangzhou 310027, China;
    kamran2093@yahoo.com
6   Department of Agronomy, Faculty of Agriculture, University of Poonch Rawalakot,
    Azad Kashmir 12350, Pakistan; khalidmehmoodagro@upr.edu.pk
*   Correspondence: nkohjackson@issas.ac.cn (J.N.N.); mabaquy@hstu.ac.bd (M.A.-A.B.);
    smia_agr@pstu.ac.bd (S.M.); Tel.: +86-131-2172-6308 (J.N.N.)

**Abstract:** Biochar research has experienced a significant increase in the recent two decades. It is growing quickly, with hundreds of reviews, including meta-analyses, that have been published reporting diverse effects of biochar on soil properties and plant performance. However, an in-depth synthesis of biochar–soil interactions at the molecular level is not available. For instance, in many meta-analyses, the effects of biochar on soil properties and functions were summarized without focusing on the specificity of the biochar and soil properties. When applied to soils, biochar interacts with different soil components including minerals, organic matter, gases, liquids, and nutrients, while it also changes soil microbial community structure and their occurrence. These different interactions modify soil physicochemical properties with consequences for dynamic changes in nutrient availability and, thus, plant performance. This review systematically analyzed biochar effects on soil properties and functions: (a) soil physical properties; (b) chemical properties; (c) biological properties; and (d) functions (plant performance, nutrient cycling, etc.). Our synthesis revealed that the surface properties of biochar (specific surface area and charge) and its associated nutrient content determine its role in the soil. At the same time, the extent of changes depends on soil properties, suggesting that both biochar and soil properties need to be considered for harvesting benefits of biochar application. Altogether, we believe our synthesis will provide a guide for researchers and practitioners for future research as well as large-scale field applications.

**Keywords:** biochar; soil physicochemical properties; soil aggregate formation; soil aggregate stability; soil fertility; pyrolysis

## 1. Introduction

The soil environment is made up of an interactive ensemble of organic matter (OM), minerals, gases, liquids, and macro/microorganisms. The interactions of these different soil components have a modifying influence on soil physicochemical properties that tend to affect the ability of soil to function as (i) an influencer of the atmosphere via mitigation of greenhouse gas (GHG) emission, (ii) a habitat for macro/microorganisms, (iii) a water reservoir and purifier, and (iv) a medium for plant growth. In this regard, the soil can be categorized as a solid phase: minerals and OM; a solution phase: soil water; and a porous phase: soil atmosphere [1,2]. Among these functions of soil, the most dominant function is to support plant growth. However, due to global climatic change (e.g., temperature and

rainfall) and intensive cultivations, it is often challenging for the soil to carry out these services. One of the main reasons for the reduced performance of the soil is its low reactive surface which affect the cycling of nutrients.

Different strategies have been employed to increase the service-providing capacity of soils. For instance, the application of OM and, recently, pyrogenic carbon in the form of biochar has been suggested [3–6]. Moreover, other problem-specific management practices (e.g., liming for acid soils and gypsum for saline soils) have also been practiced [7]. Among these amendments, biochar, the product of thermal conversion of biomass to pyrogenic carbon, has been receiving quite some interest due to multiple reasons, including its role in climate change mitigation, improvement of soil OM, and changing soil properties for a long term basis.

Hundreds of reviews have been published focusing on particular aspects of biochar and its effects on soil properties and plant performance. For instance, Natasha et al. [8] focused on the effect of biochar on uptake of trace elements, their toxicity, and detoxification mechanisms in plants. Oni et al. [9] reviewed the application of biochar to soil for remediation and its significance in the economy, while Mandal et al. [10] analyzed the properties of biochar composites and their applications to soils. Other researchers focused on (a) the application of biochar in pollution remediation [11,12], (b) the effect of biochar on soil physicochemical properties [13], and (c) the application of biochar in the management of soil acidity [14,15]. However, in many of these reviews, the mechanistic understanding between biochar-induced changes in soil properties and soil functions is not well discussed. We acknowledge that these reviews present different aspects of biochar and set a stage for a mechanistic review that can link these different aspects together. Hence, in this review, we present a critical and systematic discussion on mechanistic understanding of biochar properties and how that properties mediate soil properties and how they relate to soil functions and services.

## 2. Data Collection and Synthesis

For this review, we performed a serial search on the Web of Science (WOS) database with different keywords from July 2020 to July 2021 (Figure 1, see Supplementary materials, Text S2 for reasons why WOS was considered for this study). We searched for research and review articles between 2000 and 2021 with the search options set to all databases and titles. Since our focus was on any article containing the word "biochar", we searched for variations of the word (biochar or bio-char). Some of the searched keywords include (i) biochar or bio-char or char, or charcoal, or agrichar, (ii) biochar or bio-char + soil, (iii) biochar or bio-char + fertilizer, (iv) biochar or bio-char + agriculture, (v) biochar or bio-char + phosphate, and (vi) biochar or bio-char + greenhouse gases. The selected publications were further screened for those reporting on the (a) production conditions and chemical properties of biochar, (b) application of biochar to soils, (c) reported negative effects of biochar, (d) application of biochar to mitigate climate change, (e) effect of biochar on soil aggregation, and (f) economic importance of biochar. After all data and selected parameters were arranged as required, the effect of biochar on different soil physicochemical parameters was estimated as the percentage (%) change between the biochar amended and unamended soils. Figure 1 summarizes the search keywords and research progress with biochar between 2000 and 2021, estimated by the number of publications.

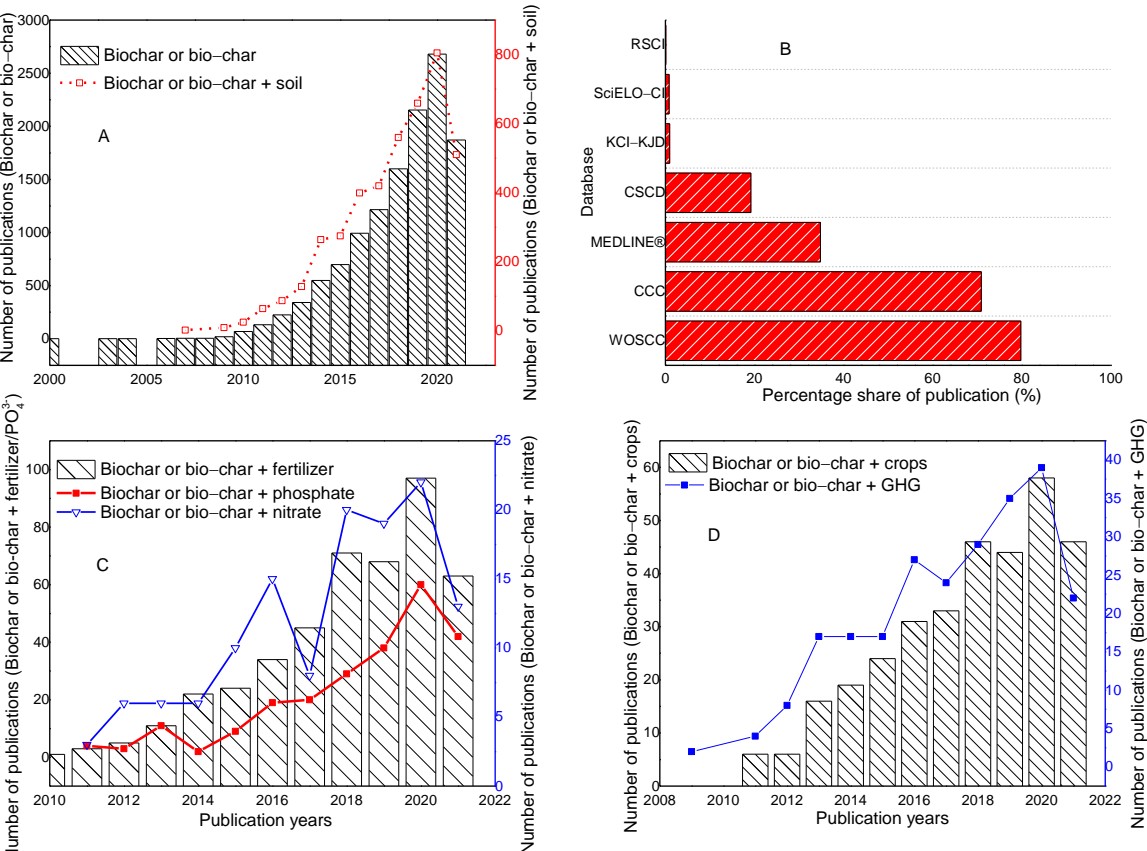

**Figure 1.** Total number of the publication containing the keywords biochar or bio−char and biochar or bio−char + soil (**A**), number of publications in different databases (**B**), biochar or bio−char + phosphate/nitrate (**C**), and biochar or bio−char + crops/greenhouse gas (GHG) (**D**). Web of Science Core Collection (WOSCC), Current Contents Connect (CCC), Russian Science Citation Index (RSCI), Chinese Science Citation Database (CSCD), KCI−Korean Journal Database (KJD).

## 3. Biochar

### 3.1. Background

Biochar (or bio-char) is a word derived by combining biomass and charcoal to denote the end product of the thermal conversion of biomass in the presence of limited or no supply of oxygen, a process commonly known as pyrolysis. It is synonymous with char, charcoal, and agrichar, although it is objectively produced for soil application to achieve soil improvements and environmental gains. Biochar can be produced using many different ways from slow to fast pyrolysis, gasification, torrefaction, hydrothermal carbonization, etc., with variable feedstock under diverse conditions (pyrolysis temperature, pressure, and residence time).

When biomass is subjected to different pyrolysis conditions, it experiences different chemical transformations: de-hydroxylation, dehydrogenation, oxidation, de-methylation, and decarboxylation. Consequently, aliphatic C chains of the biomass are converted to aromatic C. In further transformations, the aromatic C can undergo aromatic condensation to form large aromatic clusters containing fixed C, which can be connected to other units or clusters through aliphatic or aromatic side chains (Figure 2). Like many chemical processes, the chemical transformation of biomass is often accompanied by the production of intermediate products (e.g., volatile C), which could be trapped within the large aromatic clusters. Another important outcome of the biomass transformation process is the production of ash, which is the form in which nutrients in the original biomass are preserved in biochar. Thus, the produced biochar is majorly characterized to have three main contents: fixed C, volatile C, and ash. Nevertheless, the extent of these chemical processes is largely controlled by

the production conditions and feedstock type and the products formed can range from partially charred to highly carbonized materials.

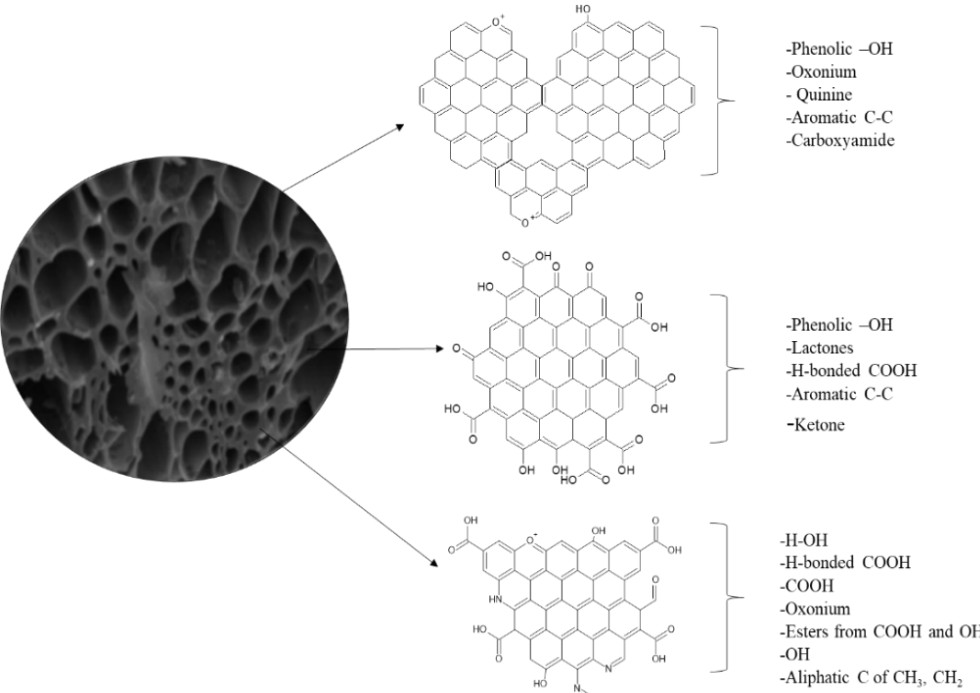

**Figure 2.** A model biochar structure with different functional groups.

### 3.2. Biochar Production Conditions and Selected Physicochemical Properties

Biochar production rate and its quality (e.g., physicochemical properties) generally depend on biomass source (feedstock), pyrolysis temperature, and pyrolysis duration [16]. Given that there is a large diversity in feedstock and production conditions, the physicochemical properties of biochar are also diverse (Table 1 Panel A, Table S1). Generally, the production rate reduces with the increase of temperature while the relative content of carbon (C) increases, but hydrogen (H) and oxygen (O) content decreases (Figure 2 and Table S1) [17]. These changes occur due to the chemical transformation of aliphatic C to aromatic C. The amount of biochar produced is reduced at higher temperatures, the nutrients in the original biomass concentrate, and thus, the ash content increases. The pH of biochar is positively associated with pyrolysis temperature due to the (a) production of alkaline compounds, (b) separation of alkali (Na, Mg, Ca, and K) salts from organic compounds, and (c) destruction of acidic functional groups [5]. Thus, the increased concentration of base cations in biochar at higher pyrolysis temperature implies that the alkalinity of biochar directly and positively relates to the total equivalent base cations [18].

Aromatic condensation during the transformation of biomass to biochar increases the specific surface area (SSA) of biochar. For example, peanut shell-derived biochar produced at 300 °C has a surface area of 3.1 $m^2$ $g^{-1}$ instead of 448.2 $m^2$ $g^{-1}$ when produced at 700 °C [19]. However, biochar loses its oxygen-containing functional groups during aromatic condensation, suggesting that low-temperature biochar generally carries a negative surface charge (CEC) [20]. On the other hand, water-extractable OM available nutrients, volatile matter content are negatively related to pyrolysis temperature and pyrolysis time [21]. Nevertheless, these relationships depend on the feedstock type, given that different biomasses present different initial carbon matrices. For instance, woody biomass was reported to have larger surface areas than grasses and chicken manure [22]. Moreover, the CEC of canola straw biochar was 199, 210, and 179 $cmol_+$ $kg^{-1}$, while that of maize biochar was 183, 304, and 210 $cmol_+$ $kg^{-1}$ when produced at 300, 500, and 700 °C, respectively [23].

Table 1 Panel B (Table S2) shows other properties of biochar such as alkalinity, CEC, exchangeable base cations ($Ca^{2+}$, $Mg^{2+}$, $Na^+$, and $K^+$), and types of functional groups. Average across all production conditions, it can be seen that biochar is generally basic while negatively charged functional groups dominate its surfaces. However, biochar may also carry a slightly positive charge if it carries sufficient positively charged functional groups (e.g., oxonium groups). Nevertheless, these surface charges are pH-dependent, suggesting that a single biochar can act as both a positive and negative surface based on the pH of the medium. Therefore, biochar generally can buffer soil pH and increase cationic retention in the soil [5,24,25]. Altogether, there are trade-offs in obtaining desired properties and therefore, a choice needs to be made for getting a certain combination of properties.

**Table 1.** The mean values of the pyrolysis temperature and physicochemical characteristics of biochar produced from different feedstock. Panel A reports the mean values of 171 studies (Table S1), while Panel B reports the mean values of 33 studies (Table S2).

| Parameter | Mean | Minimum | Maximum | SD |
|---|---|---|---|---|
| **Panel A: n = 171** | | | | |
| Pyrolysis temperature (°C) | 484.07 | 100 | 900 | 171 |
| pH | 8.84 | 3.3 | 12.4 | 1.86 |
| Yield (%) | 41.09 | 21.6 | 99.9 | 17.26 |
| Ash (%) | 16.99 | 0.1 | 81.7 | 18.70 |
| Surface area ($m^2/g$) | 94.33 | 0.76 | 907.4 | 158 |
| C (%) | 62.22 | 7.9 | 94.2 | 19.43 |
| H (%) | 3.06 | 0.3 | 25.1 | 2.46 |
| O (%) | 17.69 | 1 | 59 | 11.49 |
| N (%) | 1.39 | 0.06 | 16.6 | 1.35 |
| **Panel B: n = 33** | | | | |
| Pyrolysis temperature (°C) | 419 | 300 | 700 | 123 |
| pH | 9.25 | 6.42 | 11.32 | 1.45 |
| Alkalinity ($cmol_+$ $kg^{-1}$) | 199.1 | 79.8 | 326.1 | 76.3 |
| CEC ($cmol_+$ $kg^{-1}$) | 159.5 | 15 | 304 | 67.4 |
| Functional groups ($cmol$ $kg^{-1}$) | | | | |
| Phenolic | 99.61 | 26 | 160 | 44.1 |
| Lactonic | 34.18 | 15.6 | 51 | 9.19 |
| Carboxylic | 19.86 | 1.1 | 63.5 | 21.5 |
| Sum of exchangeable base cations ($cmol_c$ $kg^{-1}$) | 221.1 | 70.8 | 524 | 115 |

### 3.3. Preparing Designer Biochar

Production and use of function-specific biochars (e.g., adsorbent of ionic nutrients and contaminants) has been gaining attention and, therefore, engineered biochar production has increased although their large-scale application has not yet been made possible. Biochars can be modified to make their surfaces more/less positive/negative or increase their SSA. This is particularly important because an improvement in these properties has been shown to increase the efficacy of biochar as an adsorbent material. Specifically, oxidation of biochar using different methods (physical, chemical, and biological) are applied for creating negatively charged functional groups (e.g., $COO^-$) that contribute to the CEC development. Among the oxidation methods, chemical oxidation (with nitric acid, hydrogen peroxide, etc.) and biological oxidation have been reported [26–28], and the former has been shown to be more effective while the latter is easier to follow. In contrast, positive biochar properties can also be achieved with the addition of metal salts (e.g., $MgCl_2$, $AlCl_3$, $FeCl_3$) and nitrogen [29,30] since minerals are impregnated into the biochar matrices [31]. Biochar-based composite materials with function-specific surfaces have also been tailored using amine or clay minerals [32].

## 4. Biochar Modifies Soil Physicochemical Properties

### 4.1. The Effect of Biochar on Soil Hydraulic Properties and Water Holding Capacity

Biochar is a porous material and, due to its high porosity, biochar can modify soil porosity and water holding capacity (WHC). This has instigated the use of biochar in agricultural practices to mitigate the adverse effects of drought and improve soil fertility [33,34]. The ability of different biochars to modify soil physical properties is dependent on their high porosity and large surface area. This suggests that biochars with high porosity and surface area may induce the greatest effects on influencing soil physical properties. Nevertheless, porosity and surface area of biochar is largely influenced by the pyrolysis temperature and feedstock type (Table S1); therefore, high-quality feedstock and a median temperature must be selected to ensure a balance between biochar yield and desired characteristics.

Long-term (eight years) field experiments showed that even though biochar failed to improve soil aggregation or aggregate stability, it decreased soil bulk density and increased total soil porosity and macroporosity when applied at a high dose of 9 t ha$^{-1}$ y$^{-1}$ [35]. According to the authors, applying biochar to soils in high doses can significantly improve water retention in soil (including gravitational, capillary, and hygroscopic water) as well as increase plant-available water content (17.8%). However, when the biochar dosage was halved, only the plant-available water content increased (10.1%). When applied to soils, biochar decreases the soil bulk density while improving the porosity [36,37]. By decreasing soil bulk density and increasing porosity, biochar can increase soil hydraulic conductivity and water retention [35,36]. Contrary to these results, it was observed that applying finer biochar particles to soils decreased the saturated hydraulic conductivity [38]. Moreover, similar negative effects of biochar on soil saturated hydraulic conductivity were reported for coarse-grain soils [39,40]. These differences in the effect of biochar reported may be due to differences in soil types used in each study as well as on the biochar particle size given that finer biochar particles will form smaller pore sizes compared to large particles.

On comparing the effects of switchblade grass (SGB) and hemlock (HB) biochars on the WHC of a loamy sand soil, Yu et al. [41] observed that the WHC was increased by 228% and 133% when the soil was treated with 10% SGB and HB, respectively. According to the authors, the increase in WHC corresponded to 448.7% and 268.3% of the weights of SGB and HB, respectively, suggesting that SGB biochar is a better option for amending sandy soils. In a laboratory column experiment, Verheijen et al. [42] observed that amending sandy and sandy loam soils with biochar (1, 5, 10, and 20%) produced at 620 °C decreased soil bulk density while increasing the WHC. Importantly, the sandy soil experienced significant changes in bulk density and WHC even at 1% biochar, whereas the sandy loam soil only showed significant changes when biochar was applied at 5%. Further, the authors reported that the effect of biochar particle size on WHC varied for each soil type and with biochar application rate. At an application rate of 20%, WHC was increased by 53.3% and 43.1% (sandy soil) and 62.1% and 37.1% (sandy loam soil) for small (0.05–1.0 mm) and large particle (2.0–4.0 mm) size biochar, respectively [42]. For the same soil, biochar application rate has a significant correlation with soil WHC (Figure S2). Nevertheless, the magnitude of this correlation may differ with biochar type and soil type.

### 4.2. Impact of Biochar on Soil pH, Cation Exchange Capacity (CEC), and pH Buffering Capacity (pHBC)

Biochar is an alkaline material whose alkalinity depends on the ash content of the feedstock, organic and inorganic alkalis, carbonates, and functional groups [23,24]. According to evidence from 71 studies (Table 2 Panel A, Supplementary Table S4), the effect of biochar on soil pH varies with soil type, initial soil pH, and biochar application rate. The alkalinity of biochar has a strong correlation with total equivalent base cations ($R^2$ = 0.84) [18] and when applied to soils, the alkaline substances in biochar interact with the soil minerals to increase soil pH. Moreover, the functional groups on biochar surface (i.e., CEC) interact with soil active acidity and buffer soil pH while changing in nutrient dynamics can also counteract against acidification (e.g., retention and uptake of $NH_4^+$ vs conversation of

$NH_4^+$ to $NO_3^-$ and its leaching). For the same soil, the increment in pH induced by biochar increases with the application rate. The increase in pH is more significant for soils with higher initial pH and biochar with higher alkalinity (Table S3). Shi et al. [6] observed that soils with lower CEC and pHBC are highly sensitive to alkalis inputs and would respond favorably when amended with biochar relative to soils with high CEC and pHBC. Regarding the relationship between soil pH and biochar's alkalinity and pH, it was reported that biochar's alkalinity, and not pH, makes the most significant contribution ($R^2 = 0.95$) in enhancing soil pH [43].

**Table 2.** The effect of biochar application on soil physicochemical (Panel A, Table S3) and exchangeable (Panel B, Table S4) properties.

| Parameter | Mean | Minimum | Maximum | SD |
|---|---|---|---|---|
| **Panel A: n = 71** | | | | |
| Pyrolysis temperature (°C) | 368 | 300 | 400 | 27.6 |
| Soil pH | 5.61 | 3.99 | 8.40 | 1.06 |
| $\Delta$pH due to biochar | 1.12 | 0.01 | 3.44 | 0.79 |
| CEC (mmol kg$^{-1}$) | 91.5 | 51.5 | 177.2 | 27.3 |
| % increase in CEC due to biochar | 18.6 | $-17.2$ | 82.8 | 21.0 |
| pHBC (mmol kg$^{-1}$ pH$^{-1}$) | 26.0 | 12.0 | 41.7 | 7.60 |
| % increase in pHBC due to biochar | 52.0 | 1.02 | 198.5 | 45.0 |
| **Panel B: n = 54** | | | | |
| Soil pH | 4.81 | 3.99 | 5.97 | 0.44 |
| $\Delta$pH due to biochar | 0.57 | 0.01 | 1.53 | 0.37 |
| Exchangeable acidity (mmol$_+$ kg$^{-1}$) | 30.9 | 0.9 | 70.2 | 17.2 |
| % decrease in exchangeable acidity due to biochar | 49.2 | 8.57 | 96.6 | 24.5 |
| Exchangeable Al (mmol$_+$ kg$^{-1}$) | 29.2 | 0.9 | 67.7 | 16.7 |
| % decrease in exchangeable Al due to biochar | 48.5 | 0.7 | 96.5 | 25.1 |
| Exchangeable base cations (mmol$_+$ kg$^{-1}$) | 64.8 | 8.8 | 118.6 | 25.7 |
| % increase in exchangeable base cations due to biochar | 95.0 | 16.2 | 243.5 | 57.3 |

The CEC is a measure of a soil's ability to retain nutrients ($H^+$, $Ca^{2+}$, $Mg^{2+}$, $Na^+$, or $NH_4^+$), and it plays the important role of highlighting how fertile soils are [44]. The O-containing functional groups on biochar surface can deprotonate/protonate at different pH values, thereby making their surface charge characteristics pH-dependent, which becomes progressively negative with increasing pH [43,45,46]. After amending soils with biochar, the CEC of the soils is enhanced given that functional groups of biochar are added to the soil reactive surfaces [5,25,47]. However, the extent of change in soil CEC depends on the relative contribution of biochar (CEC of biochar and its application rates), with an estimated increase of up to 82.8% (applied at 5%) (Table 2 and Supplementary Table S3). In contrast, negative effects of $-17.2\%$ (applied at 1%) have also been reported (Tables 2 and S3).

Soil pHBC measures how much acid or alkali is required to decrease or increase the pH of soil by one unit [25]. After amending soils with biochar, the exchangeability of the soil was enhanced, given that biochar has abundant anionic functional groups (e.g., $COO^-$) and cation exchange sites to accommodate $H^+$ [5,25,47]. Importantly, the ability of biochar to enhance the soil exchange capacity is a direct result of its enhancing effect on soil CEC. As shown in Table S3, the effect of biochar on soil CEC and pHBC increases with biochar application rate, and the greater its effect on CEC, the greater is its effect on pHBC. Specifically, at application rates of 3% and 5%, peanut straw biochar increased the CEC of Ultisol derived from granite by 47.2% and 82.8% while increasing its pHBC by 65.8% and 123.2%, respectively. Similarly, at the same rates and for an Oxisol from basalt, the same biochar increased the CEC by 34.2% and 51.3%, while the pHBC was increased by 46.3% and 92.0%, respectively (Table S3) [25].

Attenuated total reflectance (ATR)-FTIR spectroscopic analysis revealed that biochar's enhancing effect on pHBC is dominated by the protonation of carboxylate groups on the biochar's surface (Biochar–$COO^-$ + $H^+$ ⇌ Biochar–COOH) and the decrease in soil effective CEC (ECEC) during acidification [5,47]. The curated data show that biochars with higher alkalinity and functional groups (Table S2) tend to significantly enhance soil CEC and, consequently, pHBC compared to those with lower alkalinity and functional groups, and the trend was similar for different soils (Table S3). A plot of the % increase in CEC against the % increase in pHBC due to biochar addition (Figure 3) shows a significant linear correlation ($R^2$ = 0.7241) and supports the inference that the enhancing effect of biochar on pHBC is a consequence of its enhancing effect on soil CEC.

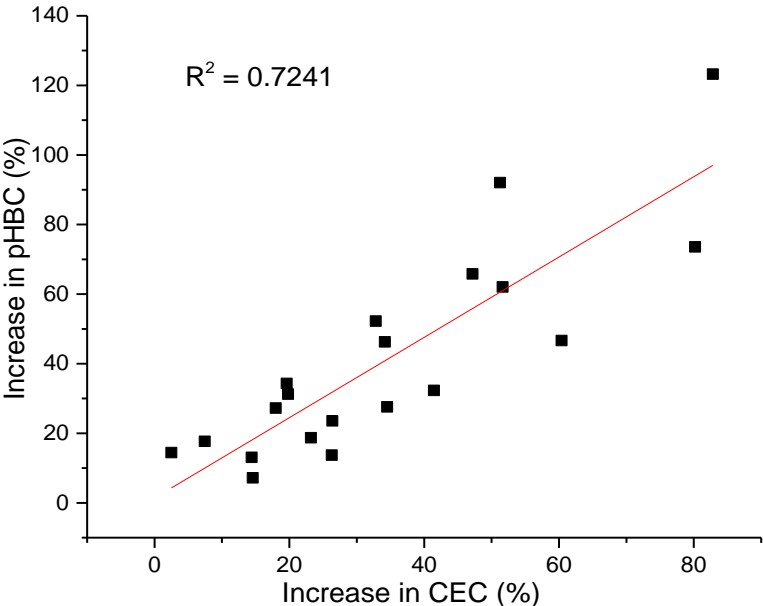

**Figure 3.** The relationship between the % increase in soil pHBC and CEC as influenced by biochar addition at various rates (n = 20). The complete data is given in Table S3 and curated from the literature [6,25].

### 4.3. Impact of Biochar on Soil Exchangeable Properties

When incorporated into soils, biochar tends to modify the physicochemical properties of soils, including their exchangeable properties. Of these exchangeable properties, the exchangeable acidity ($Al^{3+}$ + $H^+$) is of particular concern given that a significant portion (~96–100%) of it is made up of the phytotoxic exchangeable $Al^{3+}$ (Table 2 Panel B, Supplementary Table S4). The high alkalinity of biochar improves soil pH and consequently decreases exchangeable acidity, and the effect depends on the biochar feedstock, application rate, and alkalinity. For the same biochar and the same soil type, the decrease in exchangeable acidity increases with the biochar application rate (Table S4). The relationship between an increase in soil pH and a % decrease in soil exchangeable acidity induced by biochar (Figure 4a) shows a significant correlation ($R^2$ = 0.6065) and suggests that enhancing soil pH is the dominant mechanism through which biochar reduces soil exchangeable acidity.

The alkaline nature of biochar results in soils with high pH and low exchangeable acidity. Because biochar decreases exchangeable acidity by enhancing soil pH, it is expected that exchangeable $Al^{3+}$ should be affected in the same way since it is the major component of exchangeable acidity. However, the correlation between an increase in pH and a % decrease in exchangeable $Al^{3+}$ (Figure 4b) shows that this mechanism is only responsible for about 51.7% reduction in the exchangeable $Al^{3+}$ ($R^2$ = 0.5171). Nevertheless, the relationship between % decrease in exchangeable $Al^{3+}$ and % decrease in soil exchangeable acidity (Figure 4c) suggests that the decrease in exchangeable $Al^{3+}$ is predominantly due

to the decrease in exchangeable acidity ($R^2 = 0.9623$) induced by biochar and not a direct consequence of an increase in soil pH.

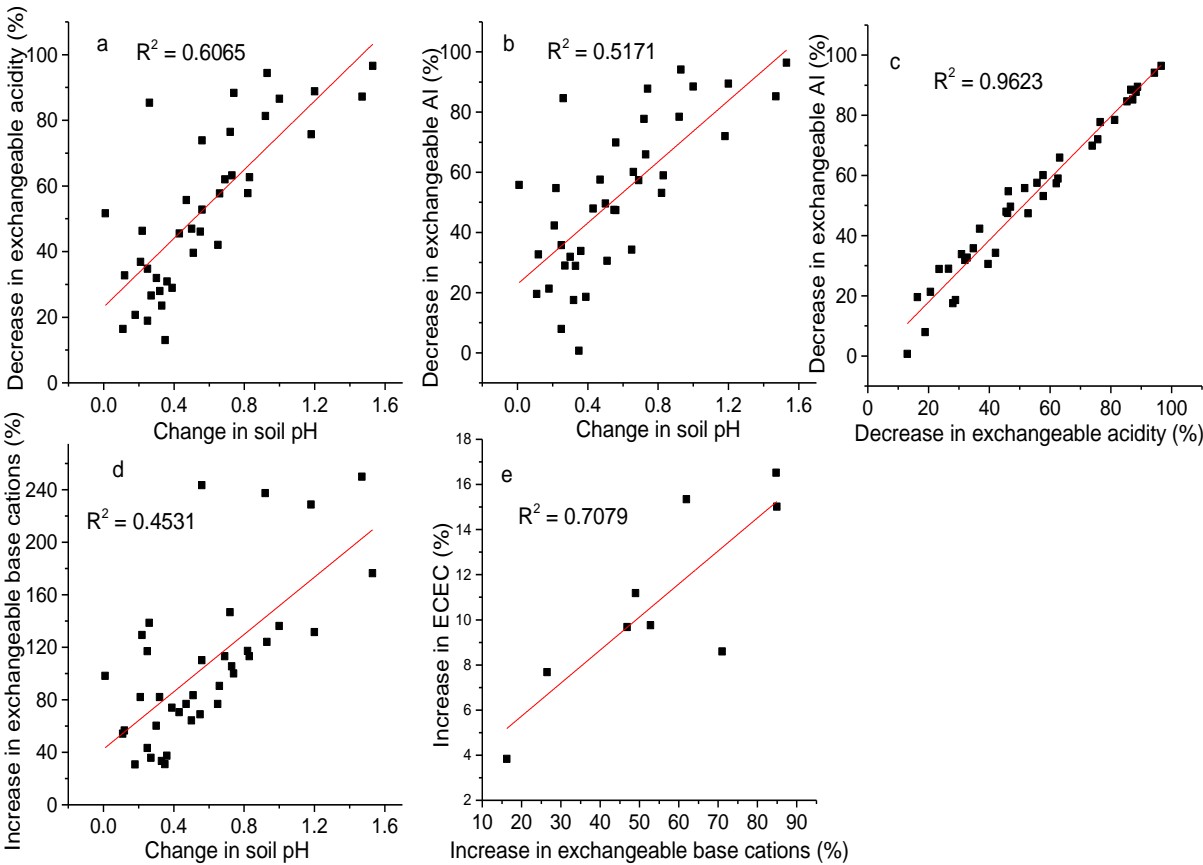

**Figure 4.** The relationship between a decrease in exchangeable acidity (**a**), exchangeable Al (**b**) and soil pH (n = 38), decrease in exchangeable acidity and exchangeable Al (**c**), exchangeable base cations ($Ca^{2+} + Mg^{2+} + K^+ + Na^+$) and soil pH ((**d**), n = 38), and effective cation exchange capacity (ECEC) and exchangeable base cations ((**e**), n = 9) as induced by biochar. The complete data is presented in Table S4.

It is true that by enhancing soil pH, biochar promotes the hydrolysis of free $Al^{3+}$ (to $Al(OH)_3$, $Al(OH)_4{}^-$, etc.) [6,48]. During this process, some of the $Al^{3+}$ can also be re-adsorbed onto the soil-biochar surface functional groups (e.g., $COO\text{-}Al^{2+}$, $\text{-}O\text{-}Al^{2+}$) to form part of the Al bound to soil organic matter (SOM). Thus, this fraction of sorbed exchangeable $Al^{3+}$ will not be accounted for by the increase in soil pH caused by biochar. By this reasoning, the relationship between soil pH and exchangeable $Al^{3+}$ as induced by biochar will always underestimate the content of $Al^{3+}$ compared to the relationship between exchangeable acidity and $Al^{3+}$. Therefore, it is recommended that soil exchangeable $Al^{3+}$ against exchangeable acidity plots be used and not against pH to estimate the effectiveness of soil amendment materials in alleviating Al toxicity in acidic soils.

Biochar contains significant amounts of exchangeable base cations ($Ca^{2+}$, $Mg^{2+}$, $K^+$, and $Na^+$) (Table S2) that can be exchanged for $Al^{3+}$ and $H^+$ when biochar is incorporated into soils. This implies that adding biochars to soils enriches nutrient cations, and the effect depends on the quantity of base cations in pristine biochar (Table S4) [4]. The relationship between exchangeable base cations and soil pH (Figure 4d) shows that pH change does not significantly ($R^2 = 0.4531$) affect the discharge of base cations from the exchange sites of biochar but, rather, it depends more on the change in effective CEC (ECEC, $R^2 = 0.7079$) (Figure 4e). This indicates that when biochar is added to soils, it increases the CEC and the overall surface negative charge, resulting in the discharge of base cations to balance the negative charge [43,47].

### 4.4. Impact of Biochar on Soil Zeta Potential

Several studies have reported on the enhancing effects of different biochars on soil CEC and ECEC. While CEC is the quantitative representation of soil negative charge at pH 7.0, the ECEC represents soil negative charge at field conditions or a given pH. Due to the complexity involved in determining both CEC and ECEC, soil scientists often evaluate soil surface charge characteristics using zeta potential measurements. The zeta potential value and sign give an idea of the charge state of a surface and represent the potential in the sliding planes of colloidal particles. The high pH of biochar renders its surface naturally negative due to the deprotonation of carboxylic acid functional groups on its surface (-COOH to –COO$^-$) [46,49], which takes part in the chemical interaction between biochar particles and soil colloids.

An important view of soil-biochar interaction is that it involves both chemical (chemical bonding) and physical interactions (e.g., electrostatic). The physical aspect of soil-biochar interaction can be estimated when exchangeable base cations (Ca$^{2+}$, Mg$^{2+}$, Na$^+$, K$^+$) and exchangeable acidity (Al$^{3+}$ and H$^+$) are released into soil solutions to form the basis for determining soil CEC. Given that the extracted exchangeable base cations and exchangeable acidity are located in the diffuse layer of an electrical double layer of soil colloids, they are considered electrostatically bonded to soil-biochar composites and can therefore not alter the overall charge behavior of soil-biochar composites [43,47]. Contrary to this, the interactions between biochar functional groups (e.g., –COO$^-$) and soil minerals (e.g., Fe/Al oxides) occur via chemical bonding (specific adsorption) to form composites with bonds of the form biochar–carboxyl–iron/aluminum–soil (biochar-CO-O-Fe/Al-soil). This kind of specific adsorption can be estimated by evaluating the changes in colloidal surface charge characteristics given that the adsorbed molecule (e.g., –COO$^-$) has to migrate to the Stern layer of the electrical double layers to form chemical bonds [50]. When this occurs, the negative charge of biochar is transferred to soil-biochar composites to alter the surface charge characteristic of the soil. Hence, zeta potential measurements have shown an increase in the negative zeta potentials of different soils treated with biochar, and the magnitude of zeta potential change was proportional to biochar application rates and content of biochar functional groups [43,45,46,49].

## 5. The Influence of Biochar on Soil Aggregation

### 5.1. The Role of Biochar in Soil Aggregate Formation

Soil aggregates are soil clumps whose sizes vary from micro (<0.25 mm) to macro-aggregates (<0.25 to >0.25 mm in diameter). A soil aggregate is formed through the binding of soil particles with cementing agents released by soil microorganisms, and it has been shown to significantly influence soil functions, including nutrient availability, water, and air movement, and soil fertility [1]. The formation of soil aggregates occurs with complex dynamic interactions of soil minerals, soil particle size (i.e., texture), inorganic binding agents (e.g., Mg, oxyhydroxides), biological binding agents (e.g., soil organic carbon, glomalin-related soil protein, microbial biomass carbon), and biological activities [51]. These aggregates often form different shapes (e.g., granular, platy, and blocky) and create pore spaces that facilitate the movement of air and water needed for healthy plant growth. The SOM greatly regulates the formation of soil structure and the addition of biochar to soils changes soil carbon sequestration by changing SOM stabilization through organo-mineral complexation [13,52].

Being a diverse material, biochar can have variable effects on soil physical properties, including aggregate formation and stability [53,54]. The ability of biochar to modify the physical and chemical properties of soils is important for amending highly weathered soils and retarding soil degradation associated with erosion [53]. Nevertheless, the ability of biochar to promote aggregation is largely influenced by biochar feedstock, biochar production condition, and soil type [55,56]. This means that while biochar can favor aggregate formation in some soils, it can inhibit its formation in others. For instance, Blanco-Canqui [57] reported a −15–226% change in soil aggregate formation using data

of 34 soils (Figure S1). Their result suggests that biochar could have variable effects on soil aggregation and stability and may only be resolved with a proper mechanistic understanding of the interactions between biochar and soil.

Biochar can enhance soil aggregation in multiple ways. Firstly, biochar interacts with the reactive surfaces of soil minerals through ligand exchange, electrostatic, π-π, and hydrophobic interactions which results in biochar–mineral–OM complexes (micro-aggregates) [58] (Figure 5a). Therefore, biochars with larger amounts of O-containing functional groups would accelerate the aggregation process faster than biochars with larger H content. These results suggest that biochar produced at low temperature and biochar that has sufficiently aged could be more effective in promoting aggregation due to their large content of O-containing functional groups [59,60]. Secondly, biochar contains abundant base cations (e.g., Ca, Mg) that can bridge the binding of negatively charged minerals together by getting adsorbed in the mineral surfaces; a process known as "flocculation" [61]. Thirdly, the addition of biochar to soils improves soil organic carbon (SOC), and the SOC indirectly influences aggregation by enhancing the contents of biological binding agents [58].

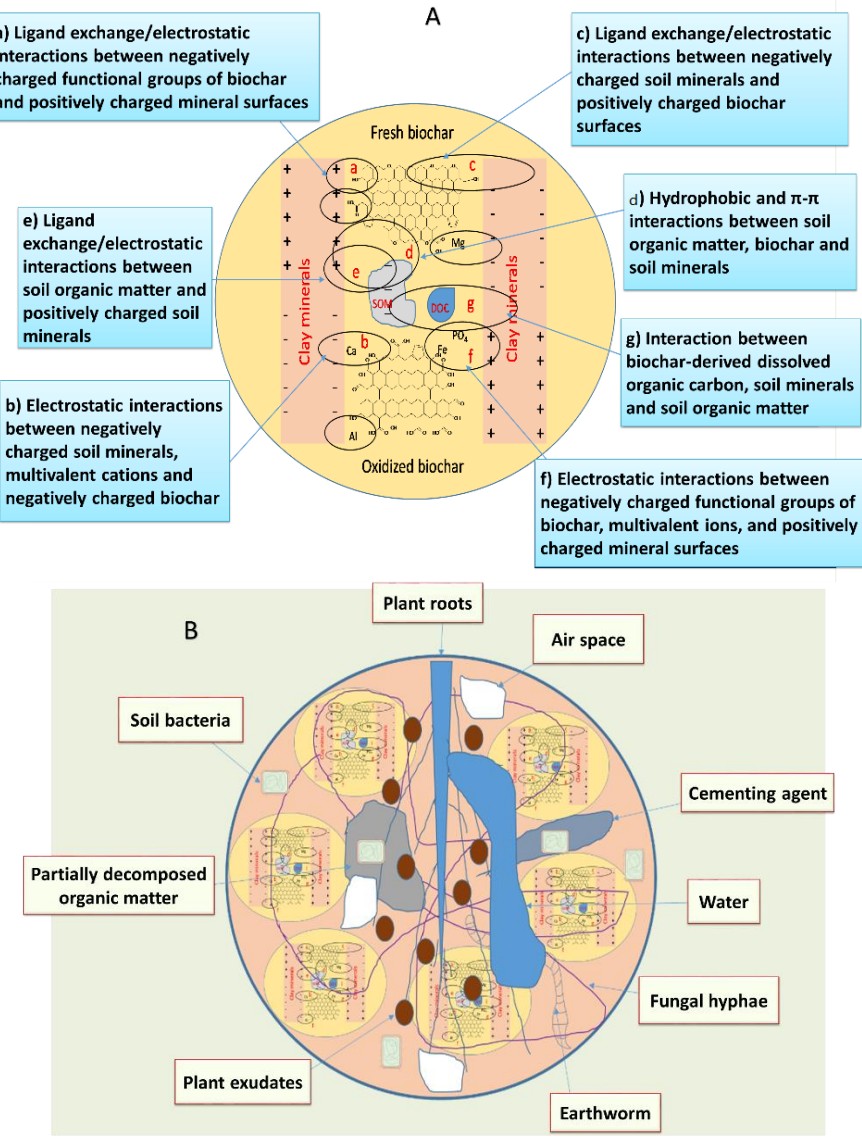

**Figure 5.** Schematic diagram of the soil micro-aggregate (**A**) and macro-aggregate (**B**) stability mechanisms by the biochar.



The formation of micro and macro-aggregates and their degradation in the soil is dynamic. It is hypothesized in this area of study by one school of thought that macro-aggregates are formed from micro-aggregates while in another school of thought, the opposite presumption is considered [62,63]. Without getting insights into the detailed mechanisms, we considered the first hypothesis for our discussion. Once the micro-aggregates are formed and biochar is applied to soils, it helps to create macro-aggregates by increasing microbial population (e.g., fungal hyphae, bacterial biomass) along with the plant-derived OM addition [64,65] (Figure 5b). Moreover, biochar is hydrophobic and can bring particles together to form aggregates, while its large surface area facilitates interactions between soil components. Thus, through these mechanisms, biochar increased macro-aggregation (>250 μm) by 25% while the micro-aggregates reduced by 7.1% [52].

*5.2. The Effect of Biochar on Soil Aggregate Stability*

The stability of soil aggregates in water is controlled mainly by internal soil forces such as van der Waals attraction, hydration, and electrostatic forces [66]. Biochar can change soil aggregate stability by altering (i) the interaction forces, (ii) the chemical interactions between its components with soil aggregates, (iii) the surface charge characteristics, (iv) the SSA and mineral addition, and (v) the microbial community dynamics [66,67]. The underlying mechanisms attributed to this increased stability are the strong interaction of biochar particles with soil minerals or microbes and an increase in electrolyte concentration. The stability of soil aggregates after biochar amendments was found to be variable. For instance, Sun and Lu [68] reported increased stability of soil aggregates after biochar application in clay-rich soil. Further, Pituello et al. [69] observed that the application of biochar to soils improves aggregate stability and soil physical fertility, with the effect being dominant for soils with a coarse texture and smaller content of organic carbon. Similar results were also reported for a range of biochars and soils [51].

Zhang et al. [70] observed in a one-year study that, irrespective of application rate, biochar addition to soils did not significantly alter soil aggregation and stability even though the SOC was improved. In a field experiment, Zhou et al. [35] studied the effect of corn cob-derived biochar on soil aggregation and aggregate stability by applying biochar at rates of 4.5 and 9 t ha$^{-1}$ year$^{-1}$. After eight years of application, biochar did not significantly affect soil aggregation or aggregate stability. In a recent study, Burgeon et al. [60] demonstrated that soil receiving fire-derived pyrogenic carbon contained more macro-aggregates than the reference soils. This suggests that pyrogenic carbon facilitated the formation of macro-aggregates through binding soils constituents and micro-aggregates together. Therefore, chemical interactions between larger size biochar, after ageing and carrying more functional groups, can form aggregates with greater stability [27]. However, the role of these diverse biochars could also depend on soil properties, including mineral composition, texture, pH, and initial SOM.

## 6. Effect of Biochar Application on Soil Biological Properties

After soil application, biochars change on soil physicochemical properties, which tend to alter the abundance of soil microbial communities [71,72]. For instance, the application of biochar enhanced soil pH and increased the content of $NH_4^+$, thereby facilitating the activities of ammonia oxidizers [73]. Amending acidic soils with biochars of rice and peanut straws increased soil pH and the abundance of ammonia-oxidizing archaea but decreased that of ammonia-oxidizing bacteria, thus, inhibiting nitrification during fertilization with urea [74]. In soils contaminated with polycyclic aromatic hydrocarbons (PAHs), using biochar for remediation has proven effective in maintaining microbial diversity [75]. The authors observed that the toxicity and bioavailability of the pollutants were significantly reduced while the growth of rare bacterial genera was enhanced. Kim et al. [76] observed that biochar application to soils increased the diversity of *terra preta* by up to 25% while Xu et al. [77] observed a significant increase in the abundance of α-diversity when an acidic soil (pH 4.48) was amended with biochar. According to Chen et al. [78], the effect

of biochar amendment on microbial community structure varied for different paddy soils. For instance, when paddy soil from Jiangxi province in China was amended with biochar at rates of 20/40 t ha$^{-1}$, the relative abundances of *Betaproteobacteria* and *Deltaproteobacteria* were increased by 54%/80% and 164%/151%, respectively. Contrarily, a paddy soil from Sichuan province amended with similar biochar (at same rates) demonstrated a decrease in the abundance of *Betaproteobacteria* by 46%/52% while it increased that of *Chloroflexi* by 27%/61% [78].

The common practice of disposing of industrial and municipal waste through landfills has potential adverse effects on soil microbial life. In their study, Lu et al. [79] demonstrated that long-term biochar amendment (three years) in the topsoil of a subtropical landfill cover enriched the abundance of *Proteobacteria* and *Acidobacteria* but not that of *Proteobacteria* and *Acidobacteria*. According to the authors, biochar-amended soils had a decreased abundance of functional genes associated with C/N cycling while those associated with P cycling increased. Nevertheless, the effect of biochar amendment on soil microbial community composition varies with biochar feedstock [80]. In their study, Yu et al. [72] showed that after amending an Oxisol and Mollisol with biochar, soil type was the predominant determining factor of microbial community composition during the initial stages of the experiment and followed by the cropping system. Moreover, it was suggested that biochar mineralization may be a biological process, and accordingly, biochar C availability and large surface area were the main factors influencing microbial colonization [81].

## 7. Implications of Biochar Interactions with Soils for Agricultural Productivity

The physicochemical properties of soils are important parameters that enable soil scientists to understand how soils change over time. Soil properties such as pH, CEC, and anion exchange capacity play important roles in evaluating the health and fertility of soils [82]. When altered, these parameters can be used as indicators for deciding what treatment is required for what soil. The sole application of biochar to soil is sometimes not sufficient to supply the required amount of essential nutrients to plants. The application of biochar along with vermicompost or compost has been suggested to enhance soil fertility and productivity [83]. On the other hand, quick decomposition of organic materials promotes nutrient loss and may pose serious problems on nutrient pollution in aquatic ecosystems. Thus, co-application of compost and biochar is proposed as an option to reduce potential environmental risks [84]. Figure S3 depicts the generalized effect of biochar on nutrient dynamics, management of soil acidification, changes in crop performance, and immobilization of pollutants, and will guide our discussion in this section.

### 7.1. Management of Acid Soils

Acid soils are characterized by low pH, low CEC, and low pHBC. These soils are prone to Al toxicity which inhibits crop growth [85–87]. Since acid soils are easily re-acidified after mineral lime application, biochar is a better alternative amendment material for correcting soil acidity due to its high alkalinity and long-lasting effect on soils [15,24]. Carbonates and surface organic functional groups are the predominant alkaline substances in biochars. When biochars are applied to acid soils, carbonates neutralize the soil acidity while organic functional groups combine with H$^+$ and Al$^{3+}$ [5]. Therefore, the application of biochars to these soils increased their pH, decreased the exchangeable acidity (Al$^{3+}$ + H$^+$), and improved the content of nutrient elements (See Supplementary Text S2, Tables S3 and S5). Adding biochar to soils enhances their exchange ability and in a laboratory study, Shi et al. [85] showed that biochar promoted root elongation in maize plants by inhibiting Al toxicity. In a four-year field experiment, it was observed that biochars significantly increased canola seed and straw yield from four crops due to their ameliorating effect on soil acidity and Al toxicity [86]. From a global-scale meta-analysis, biochar elicited a 25% average crop yield increment in the tropics, with little effect in the temperate zones [88]. This observation is reasonable given that most soils in the tropics are acidic compared to temperate zones, and when added to acidic soils, biochars can exchange intrinsic nutrient

cations (e.g., $Ca^{2+}$, $Mg^{2+}$, $K^+$) for $H^+$ in these soils. This results in an increase in soil pH and bioavailable nutrients required for crop growth.

### 7.2. Management of Alkaline Soils

Given the high intrinsic pH and alkalinity of biochar, it is not commonly used in alkaline soils (pH $\geq$ 7.4) since it may negatively affect the quality of the soil [89]. In combination with NPK fertilizer, biochar application to alkaline soils could be a viable strategy to decrease soil pH and maintain a high nutrient level [90]. The high nutrient level played an important role in improving seed yield and total biomass of soybean grown on alkaline soils. Even though biochar negatively affected soil bulk density, its application to alkaline soil significantly improved soil CEC, nitrate retention, OC, and bioavailable K [91]. Negatively, because of adsorption onto the surface of biochar, the addition of biochar to alkaline soils resulted in reduced levels of ammonium and phosphate [92,93]. The authors suggested that strong adsorption of P to the soil-biochar complex was responsible for this decrease given that the desorption of P pre-adsorbed to Al/Fe-biochar composites was slow. Nevertheless, it is recommended that acidified biochar be used for application in alkaline soils since it contains a significant amount of oxygen-containing functional groups and nutrients required for plant growth [94]. The use of nutrient-doped or nutrient-laden biochar-based fertilizers to improve the quality of alkaline soils is fast becoming popular. For instance, Elkhlifi et al. [95] and Wu et al. [96] observed that P-laden lanthanum-based biochar and iron-modified biochar improved soil P availability when applied to alkaline soils, respectively. The versatility of biochar—pristine or modified—makes it a suitable material for amending all kinds of soils, and when composted, important plant nutrients such as N, P, and K were significantly increased in alkaline soils [97].

### 7.3. Management of Saline Soils

Salinization is of growing concern, given that plant growth becomes suppressed due to osmotic imbalances and oxidative stress [98]. The management of saline soils with appropriate rates of biochar has a mitigating effect on N leaching due to enhanced retention and reduced volatilization of ammonia [99]. The ability of biochar to modify soil physicochemical properties plays an important role in microbial colonization. This has demonstrated positive effects on the abundance of ammonia-oxidizing microorganisms in alkaline soils and the decrease of N leaching [100]. In their study, Cui et al. [101] showed that a combined application of biochar with effective microorganisms decreased the impact of saline stress on the growth of *S. cannabin* by improving soil fertility and nutrient content. The increasing interest in the co-application of biochar and microorganisms for plant growth stems from the fact that biochar modifies soil physicochemical properties to improve colonization by *Arbuscular mycorrhizal* fungi (AMF) [102]. From experimental evidence, such a combination can mitigate drought-related stress on plants growth by improving the WHC [103].

Biochar has diversified mitigating impacts on salt-induced stress. Mechanistically, biochar surfaces could interact with soil constituents, including soil minerals, ions, and organic matter. Therefore, it is likely that biochar could buffer soil salt since it could provide exchange sites for ionic salts. Our understanding of how biochar surfaces could affect soil ionic salts is not complete since it is unclear how effective it would be in reducing salt stress when the ions are retained in the ion-exchange sites. In addition, biochar often carries some nutrients, including basic cations ($Ca^{2+}$, $Mg^{2+}$ and $K^+$) that could reduce salt stress. The mineral concentration in biochar is high when it is produced from manures or grasses. However, biochar application at high rates could increase salt stress by adding salts and small organic molecules. One of the most important aspects of biochar is that it is porous and, thus, carries large SSA. This large SSA contributes to buffering of salts while improving water retention in soil (Figure 6). Given that the biochar's pores are narrow, it is not certain whether the water retained in the pore would help plant water uptake.

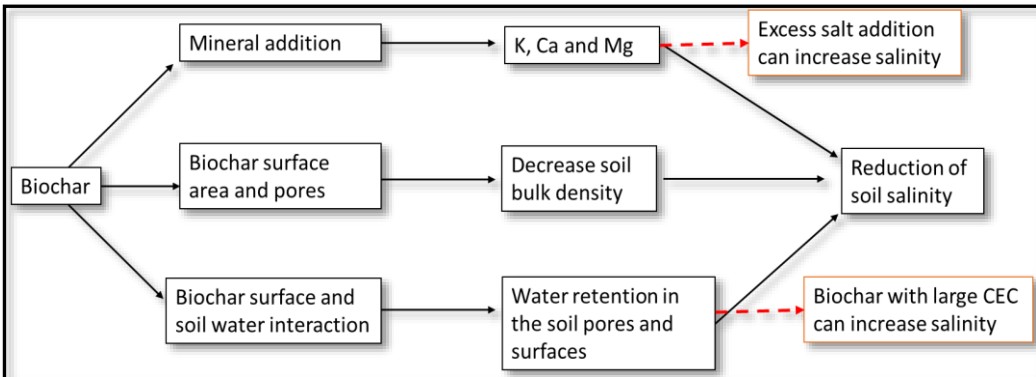

**Figure 6.** Role of biochar in the remediation of saline soils.

### 7.4. Management of Polluted Soils

The use of biochar for pollution remediation has seen a significant increase in recent times due to abundant literature attesting to the versatile nature of biochar [104,105]. As pointed out earlier, using biochar as a soil amendment material increases soil pH, CEC, and OM content. By improving these parameters, biochar indirectly boosts the ability of soils to retain pollutants such as heavy metals [106], dyes [107], antibiotics [108], etc. The charge states of both pollutants (organic and inorganic) at low soil pH make them highly available for plants and other life forms, thereby significantly increasing their toxicity potential [109]. Amending polluted soils with biochar improves pollutant retention by (i) enhancing electrostatic adsorption of cationic pollutants, (ii) introducing more O-containing organic units for complexation with or degradation of pollutants, (iii) reducing pollutants to less toxic forms, (iv) precipitating pollutants as non-toxic salts, etc. The immobilization of heavy metals such as Lead (Pb) in polluted soils was shown to follow the listed mechanisms and with the precipitation of less toxic $Pb_5(PO_4)_3Cl$, $Pb_3(CO_3)_2(OH)_2$, and $Pb(OH)_2$ [110,111]. A direct consequence of increased immobilization of pollutants in soils is reduced bioavailability. For instance, the uptake of heavy metals (Cr, Ni, Cu, Pb, Zn, and Cd) by above- and below-ground parts of rice was significantly inhibited when a paddy soil was amended with biochar [104]. Similarly, biochar amendment improved growth parameters of maize plants cultivated under Pb toxicity [112], reduced the uptake of organic pollutants by plants [113,114], enhanced adsorption and degradation of organic pollutants in soils [115], and improved microbial abundance in polluted soils [75].

### 7.5. Mitigation of Greenhouse Gas Emission

The use of pyrolysis for biochar production has been termed "a carbon-neutral process" since byproducts and emissions can be captured and none are emitted (https://biochar-us.org/biochar-slows-climate-change (accessed on 21 May 2021)). However, in most cases, half of the original carbon of the biomass can be emitted if it is not trapped. Despite this, pyrolysis and biochar are considered carbon-negative technology since this technology can reduce carbon footprint in several other ways. First, amending soils with biochar can mitigate climate change due to the recalcitrant nature of biochar carbon. The stable nature of biochar carbon implies that it can be retained in the soil for decades to centuries, if not millennia. In doing so, it can induce the stabilization of other carbon forms in the soil by accelerating organo-mineral complexation (as discussed in the previous section), thereby reducing $CO_2$ emission [116]. Across studies, it has been shown that biochar changes greenhouse gas (GHG) emission from soils through multiple ways, including (a) changes in soil nutrient dynamics, (b) changes in soil pH, EC, and Eh, and (c) changes in crop productivity, although diverse effects have also been reported (see Text S3 for details). For instance, a one-year field experiment showed that the application of NPK fertilizer to agricultural soils can increase $N_2O$ emissions. Contrarily, the application of wood residue-derived biochar to the same soil significantly inhibited $N_2O$ emission and enhanced

C storage [117]. The altering effect of biochar on GHG emission depends on biochar type and ageing conditions, soil type, the water-filled pore space [118], and its ability to change the structure of the soil microbial community [71,72]. Nevertheless, biochars have the potential to act as an excellent adsorbent for gaseous ozone and $NO_3^-$ [119], $NH_4^+$ [120], dissolved organic carbon, and other nutrients [121]. Thus, the estimation of these compounds via extraction from biochar-amended soils or evaluation of the released gases may underestimate their content.

### 7.6. Guidelines for Biochar Application

As discussed in the previous sections, biochar's interactions with soils can be quite diverse based on their properties (Figure S4). For instance, biochar with larger SSA but with neutral to positive surfaces has been shown to have minimal impacts on crop productivity when applied to clay soil. However, the carbon sequestration potential is high when biochar with a low C/H ratio is applied to clay soils. However, if biochar with large negative surfaces is applied on the same soil, the impacts may be slightly positive on crop productivity. When biochar with a large negative charge is applied to sand or loamy soil, these effects can be positive. Similar positive effects were also reported for acidic soils from the application of biochar with diverse properties [27]. With these guiding results, biochar properties can be designed for achieving specific benefits.

### 7.7. Ecotoxicology and Negative Effects of Biochar

Although biochar has the potential to mitigate climate change due to its high carbon resilience and its usefulness in amending soils to improve fertility and remediate polluted soils, hazardous compounds—with damaging effects on plants [122] and soil organisms [123–127]—may be found in biochar. Generally, the agricultural ecosystem is primarily influenced by biochar addition in three ways: contaminants in biochar, phytotoxicity, and biochar-induced changing soil environment. Contaminants such as polycyclic aromatic hydrocarbons (PAHs), polychlorinated dibenzo-p-dioxins, volatile organic compounds, and heavy metals may be present in biochar due to the presence of contaminants in the feedstock. Thus, feedstock selection for making biochar is vital to reduce the environmental risk.

The negative effect of biochar on plant growth has been reported in several studies. Some authors have reported a significant reduction in plant biomass after biochar application at a rate of 10 t ha$^{-1}$ [128], while others observed the inhibition of seed germination [129] and a reduction in seedling growth metrics [130]. The ecotoxicological effect of biochars depends on the pyrolysis temperature and feedstock type, with biochar produced at 800 °C showing increased negative effects on plants and arthropods compared to those at 400 and 600 °C [131]. Many studies have attributed the negative effect of biochar on plant growth to environmentally persistent free radicals (EPFRs) formed during pyrolysis [132,133] and to heavy metals within the biochar [130]. For instance, Liao et al. [132] observed that amending soils with biochar had an inhibitory effect on seed germination and plant growth. The authors attributed this observation to the ability of biochar to induce oxidative stress in plants. Nevertheless, the production of EPFRs can be effectively reduced by increasing the pyrolysis temperature above 500 °C [134].

## 8. Conclusions and Future Research Directions

This review presents a systematic and critical analysis of the interaction(s) of biochar with soils and its effect on soil properties and functions. The interactions are often multifaceted and depend on both biochar and soil properties. Therefore, both biochar and soil properties should be considered for harvesting the benefits of biochars. Specifically, a matrix of biochar versus soil properties can be developed for guiding the decision-makers for large-scale biochar applications. Despite the increasing number of research with biochar in the recent decade, there are few field studies with biochar. Long-term field studies are required to verify results from short-term laboratory studies so that benefits of biochar

application (e.g., climate change mitigation, immobilization of pollutants, inhibition of acidification, etc.) would be harvested at a larger scale. The specific recommendations are-

(a) The need for technological maturity: Long-term studies are required to understand the future of the currently applied biochar. Few studies have attempted to examine the future role of biochar using artificial conditions. However, real field conditions may differ and thus, the need for long-term field studies.

(b) Accreditation of biochar: Given thousands of studies reporting diverse results, it is difficult for practitioners to use biochar and, if so, determine which of the types to use. Therefore, an international body can be formed to accredit the quality of biochar. A generalized guideline can be prepared for preparing, testing, and applying biochars for achieving a target, while equal importance needs to be paid for the long-term effects of biochar because the role of currently applied biochar may be reversed in the future.

(c) Biochar-based composites: Development and application of biochar-based composite and fertilizers can be one of the new dimensions of biochar research since there is a higher chance of obtaining biochar-nutrient/contaminants interactions than their direct application to soils. However, detailed studies are required before advocating any large-scale application.

(d) Cost-effective biochar production: Research is needed to tailor technologies that can help to produce biochar at a low cost. One of the big challenges is that many large biochar production companies are struggling to sustain their business. Efforts are to be made to harvest all possible benefits, including recycling energy. Moreover, obtaining a sustainable source of biomass is needed. The use of waste biomass (municipal waste) can be an option for that. However, suitable technologies are required for handling diverse biomass.

**Supplementary Materials:** The following are available online at https://www.mdpi.com/article/10.3390/su132413726/s1, Figure S1: Changes in soil aggregation after biochar amendments; Figure S2: The relationship between increase in soil water holding capacity (WHC) and biochar application rate for a loamy sand soil; Figure S3: Schematic of biochar's role on nutrient dynamics, management of soil acidification, changes in crop performance, and pollution remediation; Figure S4: Guiding principles of biochar application to soil; Table S1: The physicochemical properties of biochar produced from various biomass precursors using varying pyrolytic conditions; Table S2: Estimated chemical properties of biochar; Table S3: Effect of biochar incorporation on soil physicochemical properties; Table S4: Effect of biochar on soil exchangeable properties; Table S5: Impact of biochar on dynamics of soil phosphorus.

**Author Contributions:** Conceptualization, J.N.N., M.A.-A.B. and S.M.; methodology, J.N.N.; software, J.N.N., M.A.-A.B. and S.M.; validation, R.X. and R.S.; formal analysis, J.N.N. and M.A.-A.B.; data curation, J.N.N., M.A.-A.B., S.M., and M.A.K.; writing—original draft preparation, J.N.N., M.A.-A.B., S.M., R.S., M.A.K., K.M. and R.X.; writing—review and editing, J.N.N., M.A.-A.B., S.M., R.S., M.A.K., K.M. and R.X.; supervision, R.X.; project administration, J.N.N.; funding acquisition, R.S. and R.X. All authors have read and agreed to the published version of the manuscript.

**Funding:** This work was supported by the National Natural Science Foundation of China (Grant No. U19A2046); the Chinese Academy of Sciences President's International Fellowship Initiative (PIFI, No. 2021PC0066); the National Natural Science Foundation of China (Grant No. 41907019); the Natural Science Foundation of Jiangsu Province, China (Grant No. BK20191103).

**Institutional Review Board Statement:** Not applicable.

**Informed Consent Statement:** Not applicable.

**Data Availability Statement:** Not applicable.

**Acknowledgments:** We acknowledge all authors whose work paved the way for this review.

**Conflicts of Interest:** The authors declare that they have no known competing financial interests or personal relationships that could have appeared to influence the work reported in this paper.

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
