# Peer review of "A Critical-Systematic Review of the Interactions of Biochar with Soils and the Observable Outcomes"

_sustainability, doi:10.3390/su132413726_

Round 1

Reviewer 1 Report

This review article by Nkoh et al. entitled “A critical systematic review of the interactions of biochar with soils and the observable outcomes”. This review article on the interactions of biochar with soils and expected outcomes can open the window of knowledge for the researchers and practitioners as future direction for developing the agriculture sector as well as reducing the global carbon emission. Overall, the review study is well organized, informative and well described research gap and future insights. The authors also explained the mechanism of biochar with soils finely with some exceptions. The references were relevant with this review.  On this basis, I would recommend this review manuscript being accepted for publication after due revision:

  1. 5 Mitigation of greenhouse emission……In this paragraph, the authors should discuss the mechanism of greenhouse emission.
  2. 2 The author can write as caption, Schematic diagram on the soil aggregate stability by the biochar. The diagram should be cleared.
  3. The authors should maintain journal guidelines such as Fig1 or Figure 1.
  4. In Figure 4, the authors can write Ca2+, Mg2+, K+ and Na+ in the figure caption.

In case of supplementary information,

Scientific name should be italic.

Good luck!

Author Response

This review article by Nkoh et al. entitled “A critical systematic review of the interactions of biochar with soils and the observable outcomes”. on the interactions of biochar with soils and expected outcomes can open the window This review article of knowledge for the researchers and practitioners as future direction for developing the agriculture sector as well as reducing the global carbon emission. Overall, the review study is well organized, informative and well described research gap and future insights. The authors also explained the mechanism of biochar with soils finely with some exceptions. The references were relevant with this review.  On this basis, I would recommend this review manuscript being accepted for publication after due revision:

We are grateful to the reviewers for making constructive critisimcs that will definitely help us to improve the article.  

Comment 1: Mitigation of greenhouse emission……In this paragraph, the authors should discuss the mechanism of greenhouse emission.

Response: Thanks for the comments. We now included a section for reflecting the mechanisms through which biochar affect greenhouse gas emission.

Comment 2: The author can write as caption, Schematic diagram on the soil aggregate stability by the biochar. The diagram should be cleared.

Response: Thanks for the comment. We have now changes the figure captions according to the style of the journal while the diagram has also been improved for better readability.

Comment 3: The authors should maintain journal guidelines such as Fig1 or Figure 1.

Response: Thank you. All the captions have been checked and corrected according to the journal guidelines.

Comment 4: In Figure 4, the authors can write Ca2+, Mg2+, K+ and Na+ in the figure caption.

Response: The caption of Figure 4 has been re-written as suggested.

Comment 5:  In case of supplementary information, scientific name should be italic.

Response: Thank you. All the scientific names have been corrected in the supplementary information.

Reviewer 2 Report

Overall, the introduction, results and discussion, as well as conclusions are properly presented, confirming the high scientific expertise of the research team. This review systematically analyzed biochar effects on soil properties and functions: a) soil physical properties; b) chemical properties; c) biological properties; and d) functions (plant performance, nutrient cycling, etc.).. This review paper is important and worth investigation and approving, however, there are some shortcomings that must be rectified. In general, important information is presented, but the structure of this ms should be reorganized.

1.Merge 4.1. Impact of biochar on soil pH into 4.2. Impact of biochar on soil CEC and pH buffering capacity (pHBC) or put pHBC to 4.1 part;

  1. 5.3. The effect of biochar on soil hydraulic properties and water holding capacity

This part is overlapped with part 4

  1. 7. Implications of biochar interactions with soils for agricultural productivity suggest delete this part or merge it into the conclusion part as the prospect analysis
  2. Introduction: The soil environment please delete the soil environment

In summary, the authors should check the whole ms carefully to avoid some format errors and also reorganize the structure of this ms.

Author Response

Overall, the introduction, results and discussion, as well as conclusions are properly presented, confirming the high scientific expertise of the research team. This review systematically analyzed biochar effects on soil properties and functions: a) soil physical properties; b) chemical properties; c) biological properties; and d) functions (plant performance, nutrient cycling, etc.).. This review paper is important and worth investigation and approving, however, there are some shortcomings that must be rectified. In general, important information is presented, but the structure of this ms should be reorganized.

We are thankful to the reviewer for careful reading of our manuscript and providing their comments.

Comment 1: Merge 4.1. Impact of biochar on soil pH into 4.2. Impact of biochar on soil CEC and pH buffering capacity (pHBC) or put pHBC to 4.1 part;

Response: Thanks for the comments and the manuscript has been changed following the reviewers comments.

Comment 2: 5.3. The effect of biochar on soil hydraulic properties and water holding capacity. This part is overlapped with part 4

Response: Following the comments of the reviewer the subsection 5.3 has been moved to subsection 4.1.  

Comment 3: 7. Implications of biochar interactions with soils for agricultural productivity suggest delete this part or merge it into the conclusion part as the prospect analysis

Response: Thanks for the comments. We think that this part carries significant importance since it discussed how biochar can change soil productivity under different conditions. Therefore, we did not make any change in this part. However, we happy to make changes if the Editor or Reviewer make specific comments on any content of this part.

Comment 4: Introduction: The soil environment please delete the soil environment

 Response: Thanks for the comments. The words have been deleted.

In summary, the authors should check the whole ms carefully to avoid some format errors and also reorganize the structure of this ms.

 Response: Thanks for the comments. We have now made changes in the text format to meet the journal requirement.  

Reviewer 3 Report

This work proposes an extensive review on the interactions of biochar with soils. As such, the matter is of interest, however the paper suffers for serious limits:

  • The topic of this review is novel, however the application proposed is not novel enough. Try to highlight and focus on the specificity of the biochar and soil properties.
  • It should be also concerned that biochar may induced adverse effects on soil from a multiangle perspective, see “Biochar impacts on soil chemical properties, greenhouse gas emissions and forage productivity: A field experiment. Science of The Total Environment, Volume 796, 20 November 2021, 148756.” Discussions should be added to clarify this problem.
  • Section “7. Implications of biochar interactions with soils for agricultural productivity“ should be in depth discussion. I suggest the following references to be considered and added in this review.

Biochar and vermicompost improve the soil properties and the yield and quality of cucumber (Cucumis sativus L.) grown in plastic shed soil continuously cropped for different years. Agriculture, Ecosystems & Environment, Volume 315, 1 August 2021, 107425

The roles of co-composted biochar (COMBI) in improving soil quality, crop productivity, and toxic metal amelioration. Journal of Environmental Management, Volume 277, 1 January 2021, 111443

  • The expression of future research directions was unfocused. Try to set this section more relevant to the topic of this review.

Once the above concerns are fully addressed, the manuscript could be accepted for publication in this journal.

Author Response

Reviewer 3:

This work proposes an extensive review on the interactions of biochar with soils. As such, the matter is of interest, however the paper suffers for serious limits:

We are grateful to the reviewer for making useful comments to improve our manuscript.

Comment 1: The topic of this review is novel, however the application proposed is not novel enough. Try to highlight and focus on the specificity of the biochar and soil properties.

Response: We are thankful to the reviewer for this comments. We are that biochar, generally, is very diverse materials with variable surface properties that affect soil properties differently. In our manuscript, we specifically highlighted which of the biochar could bring what changes in soil. For instance, the we tried to relate biochar properties with soil properties in understanding the soil aggregation and saline soil management. However, we are happy to include the suggestions of the reviewers if we receive any specific comments on a topic.

Comment 2: It should be also concerned that biochar may induce adverse effects on soil from a multiangle perspective, see “Biochar impacts on soil chemical properties, greenhouse gas emissions and forage productivity: A field experiment. Science of The Total Environment, Volume 796, 20 November 2021, 148756.” Discussions should be added to clarify this problem.

Response: Following the suggestion of the reviewer, we now included a section while the suggested articles have been used to discuss the adverse effects of biochar.

Comment 3: Section “7. Implications of biochar interactions with soils for agricultural productivity“ should be in depth discussion. I suggest the following references to be considered and added in this review. Biochar and vermicompost improve the soil properties and the yield and quality of cucumber (Cucumis sativus L.) grown in plastic shed soil continuously cropped for different years. Agriculture, Ecosystems & Environment, Volume 315, 1 August 2021, 107425; The roles of co-composted biochar (COMBI) in improving soil quality, crop productivity, and toxic metal amelioration. Journal of Environmental Management, Volume 277, 1 January 2021, 111443.

Response: Thank you for the suggestion. The references have been added to the discussion in section 7; [83] and [84].

 Comment 4: The expression of future research directions was unfocused. Try to set this section more relevant to the topic of this review.

Response: Thanks for the comments. In the section of future research direction we were trying to identify the future research needs based on the synthesis we made. We believe that our review is comprehensive and covered most of the aspects of biochar-soil-plant interactions. Given that thousands of studies have been published on biochar effects on soil properties and plant performance, it is important to develop a model/scheme that could predict biochar effects on soil properties and crop performance. We clearly mentioned that in the future research section while we also included specific points to cover the promising areas of biochar research including the examination of long-term effects of biochar.

Round 2

Reviewer 2 Report

Overall, the topic of the study fits the scope of the journal, and the study has contributed some interesting new data points. However, the language in the manuscript needs clarification and corrections. Authors should check the whole manuscript carefully.

For example, overused “the” in The soil environment”

Figure 6. Schematic of biochar’s role in the remediation of saline soils. Please put figure 6 into the middle of the page.

Rewrite the sentence “The mitigating effect of biochar on saline-induced stress is diverse and different stud-ies have observed different results on different soils.”

Please merge 7.5 and 7.7 into part 4or 6, authors should still pay attentions to the logics of the whole ms.

Author Response

Comment 1: Overall, the topic of the study fits the scope of the journal, and the study has contributed some interesting new data points. However, the language in the manuscript needs clarification and corrections. Authors should check the whole manuscript carefully.

Response: We are grateful to the reviewer for providing valuable comments to improve our manuscript. The language of the manuscript has now been thoroughly checked. 

Comment 2: For example, overused “the” in The soil environment”

Response: Suggestion is included. 

Comment 3: Figure 6. Schematic of biochar’s role in the remediation of saline soils. Please put figure 6 into the middle of the page.

Response: Thanks for the comment. The figure is formatted according to the suggestion.

Comment 4: Rewrite the sentence “The mitigating effect of biochar on saline-induced stress is diverse and different stud-ies have observed different results on different soils.”

Response: Thanks for the suggestion. We now revised the sentence for improving readability. 

Comment 5: Please merge 7.5 and 7.7 into part 4or 6, authors should still pay attentions to the logics of the whole ms.

Response: We are thankful to the reviewer for that specific comments. Without disagreeing with the proposed suggestion, we would like to underscore that sections 7.5 and 7.6 provide very different information from sections 4-6. Specifically, we discussed the effects of biochar on specific soils and their management in sections 7.5 and 7.7 while we explained biochar's effects on soil properties. Considering these points, we kept our manuscript as it was while we are happy to make changes if the reviewer/editor suggests merging sections. 

Reviewer 3 Report

For me, It is OK.

Author Response

We are grateful to the reviewer for their comments.